REGISTERED REPORT PROTOCOL

# The experiences of patients with peritoneal dialysis: A systematic review of qualitative evidence protocol

**Man Zhang**, **Chunfeng Cai***

School of Nursing, Wuhan University, Wuhan, Hubei, China

* 1660433132@qq.com

## Abstract

### Background

The incidence of end-stage renal disease (ESRD) is on the rise, it has been a major public health problem and places a significant burden on healthcare systems. Peritoneal dialysis (PD) is an increasingly popular form of renal replacement therapy. Patients undergoing dialysis treatment undergo specific pathophysiological and psychological changes. The aim of this systematic review is to investigate the experiences of ESRD patients receiving PD in order to gain deeper insights into their attitudes and beliefs towards this treatment. This will help researchers and health professionals to target interventions to improve the quality of life of PD patients.

### Design

Protocol for a qualitative systematic review.

### Methods

This systematic review protocol will follow the Joanna Briggs Institute (JBI) meta-analysis methodology. We will conduct a comprehensive search, including English and Chinese databases. It will include all published qualitative studies of patients' experiences with PD. Both English and Chinese literature will be covered. Two reviewers will independently participate in the literature selection, document selection, and data extraction process. Synthesis will be carried out through in-depth reading of the original text and subsequent creation of similar categories.

### Results

Experiencing is a complex dimension that includes physical, psychological and social aspects. This review will enable nurses to gain a deeper understanding of the feelings and beliefs of patients with PD. Our findings will provide healthcare professionals and policy makers with evidence to provide better care for patients with PD.

**Data Availability Statement:** All relevant data from this study will be made available upon study completion.

**Funding:** The author(s) received no specific funding for this work.

**Competing interests:** The authors have declared that no competing interests exist.

## Introduction

End-stage renal disease (ESRD) means permanent and irreversible loss of kidney function with a high mortality rate. In recent years, the number of ESRD patients worldwide has been increasing, representing a major public health problem and a major burden on health systems. It is the eighth leading cause of death worldwide [1–3]. As the Aging population, the prevalence and incidence of diabetes, cardiovascular disease and hypertension continue to rise, the number of ESRD patients will gradually increase. Renal replacement therapy (RRT) is the main treatment modality for ESRD patients, including kidney transplantation (KT), hemodialysis (HD) and peritoneal dialysis (PD). Previous studies have shown that compared to dialysis therapy, patients who undergo kidney transplantation will have the best quality of life [4–6]. However, due to the limited number of transplantable organs and the long waiting time, HD and PD are the only two options for many ESRD patients [7, 8]. ESRD patients are often accompanied by sarcopenia [9], frailty [10], cardiovascular and cerebrovascular diseases [11], anxiety and depression [12] and other complications, which affect the patients'physical and psychological health seriously. Dialysis patients also have to adapt to the disease by following different diets, fluid restrictions and lifestyle changes to maintain their lives [13]. Such restrictions and changes will have a major impact on patients' beliefs about the disease and their sense of personal control. Recently, peritoneal dialysis has been accepted and chosen by more and more patients because of its advantages, such as stable hemodynamics, some residual renal function, less frequent hospital visits, lower medical costs, greater autonomy and independence, and greater patient flexibility [14–16]. However, the prevalence of peritoneal dialysis (PD) therapy is very different, it is reported that 79% of the total number of ESRD patients in Hong Kong receive peritoneal dialysis [17], 6.9% in the UK, 8% in the United States, 16% in Europe and about 20% in Italy [18, 19], Although there are many benefits to PD, there's still a lot of symptom burden, such as PD-associated peritonitis [20], volume overloaded [21] and serious caregiver burden, all these will increase the physical and psychological burden of PD patients.

In order to deeply understand the real experiences, feelings, beliefs and perspectives of patients with PD, researchers mostly use qualitative research methods, including phenomenology, ethnography, focus groups, structured interviews, grounded theory and so on. However, due to differences in the focus of individual studies, the experience of researchers and the selection of research objects, there will be differences and limitations in the research findings. Therefore, conducting a systematic review of qualitative research from different backgrounds, synthesising different findings and gaining a comprehensive understanding of the experiences and attitudes of this population will be of great importance for future research and interventions.

The purpose of this study is to conduct a systematic review of qualitative research by synthesising the qualitative evidence from people with peritoneal dialysis (PD) to gain a deeper understanding of the beliefs, feelings and expectations of people with PD and to provide in-depth insights into the physical, psychological and social adjustment experiences of people with PD. This will enable health care providers, policy makers and researchers to better understand this group, improve their prognosis, value the physical and psychological experiences of patients, incorporate them into medical decision making, and take targeted action to improve health outcomes and quality of life for patients.

## Methods

### Protocol registration and reporting

This review was registered in the International Prospective Register of Systematic Reviews (PROSPERO) database (CRD:42021246590). This study will be reported in accordance with

the PRISMA-Preporting guidelines [22]. It is a guideline for reporting the synthesis of qualitative research, consisting of 17 items and grouped into 7 main domains (S1 Checklist).

## Inclusion criteria

The inclusion criteria followed the PICoS principle:

**Population.** Our study population was the patients with ESRD undergoing PD, which was diagnosed according to the Kidney Disease Improving Global Outcomes (KDIGO) [23]. Studies are eligible regardless of patient age, sex, education, marital status, race and region. And regardless of dialysis duration.

**Phenomenon of interest.** The experiences and beliefs of people with ESRD treated with PD will be explored in this review.

**Context.** This study will consider the experiences, beliefs, attitudes and perspectives of people with PD, whether they live at home, in a hospital, nursing home or other healthcare setting. And whatever their cultural background.

**Types of studies.** This review will consider all types of qualitative studies, including but not limited to phenomenology, ethnography, grounded theory, interviews, narrative research. Both Chinese and English language studies will be included.

## Data sources and search strategy

To identify all potential published and unpublished studies, we consulted with an experienced medical librarian and conducted a comprehensive literature search in electronic databases, including the following English language databases PubMed, Embase, PsycINFO, CINAHL Complete (EBSCO), Web of Science, JBI Database of Systematic Reviews, The Cochrane Library and Chinese databases such as China National Knowledge Infrastructure (CNKI), Chinese Biomedical Literature Database (CBM). The search will include three steps, the first step was to search published papers on the PubMed database, we will conduct a combination of Medical Subject Heading (MeSH) terms and text words search on the definitions of peritoneal dialysis, experience and qualitative study in the title and abstract, and the index terms used to describe the article will also be searched. All identified keywords and index entries will then be reported and used in all relevant databases for the second step. Thirdly, the references of all included studies that may be relevant to the review are examined. To avoid publication bias and ensure that the review includes as much evidence as possible, grey literature will be searched using OpenGrey. Both English and Chinese articles will be searched. The details of our search strategy are presented in S1 Appendix.

## Study selection

Once the initial search was completed, all identified studies will be uploaded into EndNote X9. After removing duplicate studies, two reviewers (MZ and CC) independently screened all titles and abstracts that met the study inclusion criteria. The titles and abstracts were read first, then the full text was read in detail and screened again after obviously irrelevant literature was excluded. After this process, both reviewers will compare their results and any disagreements will be resolved by discussion or by seeking the assistance of a third reviewer (CC).

## Assessment of methodological quality

After the study selection process, all eligible identified final studies will be critically appraised. Two independent reviewers (MZ and CC) will appraise the selected qualitative literature using the JBI-Qualitative Appraisal Instrument (JBI-QARI). The JBI-QARI is commonly used to

| JBI Qualitative Assessment and Review Instrument (QARI) | | | | |
|---|---|---|---|---|
| | Yes | No | Unclear | Not applicable |
| 1. Is there congruity between the stated philosophical perspective and the research methodology? | | | | |
| 2. Is there congruity between the research methodology and the research question or objectives? | | | | |
| 3. Is there congruity between the research methodology and the methods used to collect data? | | | | |
| 4. Is there congruity between the research methodology and the representation and analysis of data? | | | | |
| 5. Is there congruity between the research methodology and the interpretation of results? | | | | |
| 6. Is there a statement locating the researcher culturally or theoretically? | | | | |
| 7. Is the influence of the researcher on the research, and vice-versa, addressed? | | | | |
| 8. Are participant, and their voices, adequately represented? | | | | |
| 9. Is the research ethical according to current criteria or, for recent studies, and is there evidence of ethical approval by an appropriate body? | | | | |
| 10. Do the conclusions drawn in the research report flow from the analysis or interpretation, of the data? | | | | |
| Overall appraisal: Include Exclude Seek futher info | | | | |
| Comments (including reason for exclusion): | | | | |

**Fig 1. JBI critical appraisal checklist for qualitative research.**

assess the strengths and limitations of qualitative studies and consists of 10 items, all of which were rated as 'yes', 'no', 'unclear' and 'not applicable'. The critical appraisal questions are listed in Fig 1. The results are compared after the initial assessment of the article by two reviewers. If there are any disagreements, they will be resolved through discussion or by seeking the assistance of a third reviewer.

## Data extraction

Data extraction from original research is often done according to the PICOS principle. The standardised data extraction tool from JBI SUMARI is used by two independent reviewers to extract the relevant data, which mainly includes author, year, geographical location, research design, context, participant characteristics, phenomenon of interest and main findings (Fig 2).

| Modified JBI Qualitative data extraction tool | | | | | | | | | | |
|---|---|---|---|---|---|---|---|---|---|---|
| Study (Name, Year and Authors) | Phenomena of interest | Methodology | Methods | Setting | Geographical | Cultural | Participants (Age, sample) | Data analysis | Authors conclusion | Comments |
| | | | | | | | | | | |

**Fig 2. JBI QARI data extraction tool for qualitative research.**

## Data synthesis

There are several commonly used approaches to reviewing and synthesizing qualitative evidence, such as meta-ethnography [24], narrative synthesis and meta-aggregation. The qualitative research findings will be synthesised using meta-aggregation with the JBI SUMARI approach to understand the experiences of patients with peritoneal dialysis [25]. The meta-aggregative approach is sensitive to the practicality and usability of the primary author's findings, and does not seek to reinterpret those findings. Another strong feature of the meta-aggregative approach is that it seeks to implement generalisable statements in the form of recommendations to guide practitioners and policy makers [26]. There will be three steps to synthesise the findings of the original study. In the first step, the extracted findings will be assessed by the reviewers based on the degree of fit between the data and the accompanying figure, there will be three outcomes: clear, credible and unsupported. Findings rated as unsupported will be discarded, and clear and credible findings will be included in the summary. This rating is an important indicator of the reliability of primary research and provides transparency and confidence to readers and potential users of the evidence. Second, these included findings are aggregated and categorised into a set of meaningfully similar categories. Third, these categories are synthesised to produce a new comprehensive set of synthesised findings. And these findings can serve as the basis for evidence-based practice.

## Confidence in the evidence from qualitative research synthesis

The aim of reviewing the body of evidence produced by the meta-aggregation is to ensure confidence in the final synthesised findings. This allows the evidence to be used as a basis for making recommendations or informing health care practice, and to guide clinical decision making. Reliability and credibility are two main factors that influence the confidence in the results. Once the synthesis is complete, the ConQual approach is used to assess the final findings of the research synthesis [27]. The ConQual process is an overall scoring system that can be used to assess the reliability and credibility of synthesised qualitative findings according to specific scoring criteria. After scoring, this system would give an overall score of high, moderate, low or very low for the final findings (Fig 3).

## Discussion

Peritoneal dialysis is one of the treatments for ESRD; patients with PD will experience all kinds of difficulties, whether in the physical, psychological or social dimension. Understanding the experiences of people with PD is important for improving their physical and psychological health and quality of life. This systematic review is a meta-aggregation of qualitative evidence that explores in depth the experiences, attitudes and beliefs of patients with ESRD undergoing

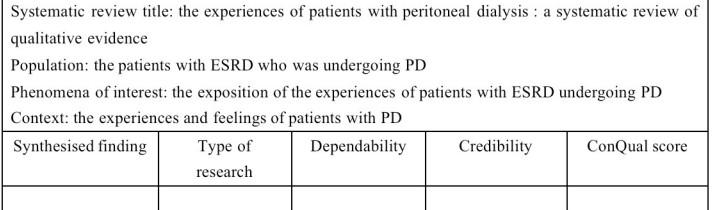

| Systematic review title: the experiences of patients with peritoneal dialysis : a systematic review of qualitative evidence | | | | |
|---|---|---|---|---|
| Population: the patients with ESRD who was undergoing PD | | | | |
| Phenomena of interest: the exposition of the experiences of patients with ESRD undergoing PD | | | | |
| Context: the experiences and feelings of patients with PD | | | | |
| Synthesised finding | Type of research | Dependability | Credibility | ConQual score |
| | | | | |

**Fig 3. ConQual summary of findings.**

PD, through the findings of this review, we can understand their physical, psychological statues and help us to learn more comprehensively about their difficulties and needs. It can also be used to inform healthcare professionals, policy makers and researchers to focus on this particular population and take targeted action in the future, particularly to enable nurses to provide comprehensive care so that patients have a better health outcome.

## Strength and limitations of this study

This systematic review is the first to systematically search both English and Chinese articles to explore the experiences of people with PD in different cultural contexts. To provide evidence for PD management and interventions in future research. Although we tried our best to perfect the whole process, there are some limitations to this review. Firstly, the definitions of experiences vary, we may not be able to describe these terms comprehensively, so the result of the search may have some weaknesses. And this review only included the articles published in English and Chinese, so this review may not be applicable to other cultural contexts.

## Supporting information

**S1 Appendix. Search strategy.**
(DOCX)

**S1 Checklist. Reporting checklist for protocol of a systematic review and meta analysis.**
(DOCX)

## Acknowledgments

We would like to thank Dr Y Wang, teacher of the library of Wuhan University School of Medicine, for advice on the search strategy and review methodology.

## Author Contributions

**Conceptualization:** Man Zhang, Chunfeng Cai.

**Data curation:** Chunfeng Cai.

**Formal analysis:** Man Zhang, Chunfeng Cai.

**Investigation:** Man Zhang.

**Methodology:** Man Zhang, Chunfeng Cai.

**Supervision:** Chunfeng Cai.

**Writing – original draft:** Man Zhang, Chunfeng Cai.

**Writing – review & editing:** Man Zhang.

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
