## [Decision Letter · Decision Letter 0]

14 Jun 2023

PONE-D-23-11383The experiences of patients with peritoneal dialysis : a systematic review of qualitative evidence protocolPLOS ONE

Dear Dr. Cai,

Thank you for submitting your manuscript to PLOS ONE. After careful consideration, we feel that it has merit but does not fully meet PLOS ONE’s publication criteria as it currently stands. Therefore, we invite you to submit a revised version of the manuscript that addresses the points raised during the review process.

We look forward to receiving your revised manuscript.

Kind regards,

Yavuz - Ayar

Academic Editor

PLOS ONE

Journal Requirements:

Additional Editor Comments:

Dear Author/s

Greetings

As a result of the evaluation made by the reviewers, the article can be published after minor revision.

Best regards

Reviewers' comments:

Reviewer's Responses to Questions

**Comments to the Author**

1. Does the manuscript provide a valid rationale for the proposed study, with clearly identified and justified research questions?

Reviewer #1: Yes

Reviewer #2: Yes

2. Is the protocol technically sound and planned in a manner that will lead to a meaningful outcome and allow testing the stated hypotheses?

Reviewer #1: Yes

Reviewer #2: Yes

3. Is the methodology feasible and described in sufficient detail to allow the work to be replicable?

Reviewer #1: Yes

Reviewer #2: Yes

4. Have the authors described where all data underlying the findings will be made available when the study is complete?

Reviewer #1: Yes

Reviewer #2: Yes

5. Is the manuscript presented in an intelligible fashion and written in standard English?

Reviewer #1: Yes

Reviewer #2: No

6. Review Comments to the Author

You may also provide optional suggestions and comments to authors that they might find helpful in planning their study.

Reviewer #1: Regarding the manuscript, in the last paragraph on page 5, you stated, "The purpose of this study is to conduct a systematic review of qualitative research by synthesising the qualitative evidence from people with Parkinson's disease (PD)." Could you please verify whether the term "Parkinson's disease (PD)" has been used inadvertently? Your clarification would be much appreciated.

Reviewer #2: Dear Editor;

The review will provide healthcare professionals and policy makers with evidence to provide better care in peritoneal dialysis patients.

My suggestion could be a contribution if it can be done:

- It can be considered if chronic PD results can be obtained in patients with hepatorenal syndrome and cardiorenal syndrome.

- Towards the end of page 5, it is referred to as parkindon disease for PD abbreviation.... let's fix it.

- In my opinion, the review plan is acceptable after correcting the native English language.

7. PLOS authors have the option to publish the peer review history of their article (what does this mean?). If published, this will include your full peer review and any attached files.

Reviewer #1: No

Reviewer #2: No

---

## [Author Response · Author response to Decision Letter 0]

24 Jun 2023

Response to Reviewers

Dear Editor and Reviewers,

First, we would like to thank you for your kind letter and for reviewers’ constructive comments concerning our artical. These comments are all valuable and helpful for improving our article. All the authors have seriously discussed about all these comments. According to the reviewers’ comments, we have modified our manuscript to meet with the requirements of your journal. In this revised version, changes to our manuscript with the document are highlighted by using red-colored text. Point-by-point responses to the reviewers are listed below in this letter.

Reviewer 1:

Comment to the author

Regarding the manuscript, in the last paragraph on page 5, you stated, "The purpose of this study is to conduct a systematic review of qualitative research by synthesising the qualitative evidence from people with Parkinson's disease (PD)." Could you please verify whether the term "Parkinson's disease (PD)" has been used inadvertently? Your clarification would be much appreciated.

Response:

We sincerely thank the reviewer for careful reading. As suggested by the reviewer, we have corrected the “Parkinson's disease” into “Peritoneal dialysis”.

Reviewer 2:

Comment 1:

It can be considered if chronic PD results can be obtained in patients with hepatorenal syndrome and cardiorenal syndrome.

Response 1:

This study focuses on the experiences of patients with end-stage renal disease treated with peritoneal dialysis to understand the internal beliefs, feelings and perceptions of peritoneal dialysis treatment in this disease population. This will provide a theoretical basis for researchers and healthcare professionals to gain a deeper and more comprehensive understanding of this population and to provide targeted interventions to improve patient outcomes. All patients undergoing peritoneal dialysis who met the inclusion and exclusion criteria will be included in this study without regard to primary medical conditions

Comment 2:

Towards the end of page 5, it is referred to as parkindon disease for PD abbreviation.... let's fix it.

Response 2:

We feel sorry for our carelessness. In our resubmitted manuscript, the typo is revised. Thanks for your correction.

Comment 3:

In my opinion, the review plan is acceptable after correcting the native English language.

Response 3:

We tried our best to improve the manuscript and made some changes to the manuscript. These changes will not influence the content and framework of the paper. And here we did not list the changes but marked in red in the revised paper. We appreciate for Editors/Reviewers’ warm work earnestly and hope that the correction will meet with approval.

---

## [Editor Report · Decision Letter 1]

4 Jul 2023

The experiences of patients with peritoneal dialysis : a systematic review of qualitative evidence protocol

PONE-D-23-11383R1

Dear Dr. Cai,

We’re pleased to inform you that your manuscript has been judged scientifically suitable for publication and will be formally accepted for publication once it meets all outstanding technical requirements.

Kind regards,

Yavuz - Ayar

Academic Editor

PLOS ONE

Additional Editor Comments (optional):

Dear Editor

Greetings

The article can be published in its current form.

Best regards
---

## [Editor Report · Acceptance letter]

10 Jul 2023

PONE-D-23-11383R1 

The experiences of patients with peritoneal dialysis : a systematic review of qualitative evidence protocol 

Dear Dr. Cai:

I'm pleased to inform you that your manuscript has been deemed suitable for publication in PLOS ONE. Congratulations! Your manuscript is now with our production department. 

Kind regards, 

on behalf of

Professor Yavuz - Ayar 

Academic Editor

PLOS ONE